# A mutational hotspot that determines highly repeatable evolution can be built and broken by silent genetic changes

James S. Horton [1✉], Louise M. Flanagan[1], Robert W. Jackson[2], Nicholas K. Priest[1,3] & Tiffany B. Taylor [1,3✉]

Mutational hotspots can determine evolutionary outcomes and make evolution repeatable. Hotspots are products of multiple evolutionary forces including mutation rate heterogeneity, but this variable is often hard to identify. In this work, we reveal that a near-deterministic genetic hotspot can be built and broken by a handful of silent mutations. We observe this when studying homologous immotile variants of the bacteria *Pseudomonas fluorescens*, AR2 and Pf0-2x. AR2 resurrects motility through highly repeatable de novo mutation of the same nucleotide in >95% lines in minimal media (*ntrB* A289C). Pf0-2x, however, evolves via a number of mutations meaning the two strains diverge significantly during adaptation. We determine that this evolutionary disparity is owed to just 6 synonymous variations within the *ntrB* locus, which we demonstrate by swapping the sites and observing that we are able to both break (>95% to 0%) and build (0% to 80%) a deterministic mutational hotspot. Our work reveals a key role for silent genetic variation in determining adaptive outcomes.

[1] Milner Centre for Evolution, Department of Biology & Biochemistry, University of Bath, Claverton Down, Bath BA2 7AY, UK. [2] School of Biosciences and Birmingham Institute of Forest Research (BIFoR), University of Birmingham, Edgbaston, Birmingham B15 2TT, UK. [3] These authors jointly supervised: Nicholas K. Priest, Tiffany B. Taylor. ✉email: j.s.horton@bath.ac.uk; t.b.taylor@bath.ac.uk

Mutational hotspots, which describe instances where independent cell lines persistently fix mutations at the same genomic sites, can make evolution remarkably repeatable. Such hotspots are of immense importance as they have been observed to drive evolution across the domains of life, from viruses (including SARS-CoV-2[1]), to bacteria (including MRSA[2]), to higher eukaryotic cell lines including those in avian species[3] and human cancers[4]. Our understanding of evolutionary dynamics (e.g. competitive selection and clonal interference) can sometimes explain the appearance of hotspots, but genetic features that build hotspots by biasing mutation rates are much less understood.

There have been many examples of experimental systems evolving via repeatable evolution. Microbes evolving under strong selection often rapidly adopt similar novel phenotypes[5,6]. Furthermore, these phenotypes are often underpinned by mutation hotspots, which come in the form of clustered genetic changes within the same region of the genome[7,8], or within limited pockets of loci[9–12]. Sometimes realised mutations are found only in genes from a single regulatory pathway[13,14] or a single protein complex[15]. In rare cases, evolutionary events can be seen to repeatedly target just a handful of sites within a single locus[16,17]. Repeatable evolution allows lines to evolve in parallel, and the degree of parallelism typically becomes less common as it descends from broader genomic regions to the nucleotide[18,19]. However, despite frequent descriptions of repeatable evolutionary events, a detailed understanding of the hotspots that ensure their occurrence is often lacking.

There are three primary facilitators of mutational hotspots that drive repeatable evolution: (i) Fixation bias, which skews evolution toward mutations that enjoy a higher likelihood of dominating the population pool. Not all facilitators of fixation bias are considered adaptively advantageous (e.g. homologous recombination events in mammalian genomes can bias gene conversion toward certain alleles[20]). But in instances where we observe rapid and highly parallel sweeps fixation bias will likely take the form of selection, which drives the fittest competing genotypes in the population to fixation[21,22]). (ii) Mutational accessibility, as there may be only a small number of readily accessible mutations a genotype can undergo to improve fitness[23]. And, (iii) Mutation bias, where genetic and molecular features scattered throughout the genome cause sites to mutate at different frequencies and toward certain mutation types (for example, A:T > G:C), constraining the mutational spectrum to favour particular outcomes[24]. Previous research shows that mutation rate heterogeneity can be influenced by the arrangement of nucleotides surrounding a particular site[25], and genetic features such as the secondary structure of DNA[26] including the formation of single-stranded DNA hairpins[27]. Nevertheless, the prominence of genetic sequence in driving parallel evolutionary outcomes remains unknown.

To establish which mechanisms are at play, it is important to consider whether parallel outcomes are robust to experimental conditions such as environment[28] and to account for clonal interference, which can alter the chance of observing parallel evolution[24,29]. Clonal interference can occur either due to standing genetic variation in the founder population which yields multiple adaptive genotypes in a novel environment (i.e. a soft selective sweep[30]) or when the rate of mutation supply is high relative to the selective coefficient[31]. Clonal interference does not often play an important role when founding experimental lines with clonal samples, performing experimental procedures over short timescales, and ensuring rapid fixation of adaptive mutants e.g. through spatial separation and/or introducing an artificial bottleneck. However, under such conditions, the primary influence of selection will manifest as clonal interference, as a large starting population may give rise to multiple adaptive genotypes which compete for fixation throughout the course of the experiment[32].

We have utilised an ideal system for identifying the key features that build mutational hotspots by employing two engineered non-flagellate and biosurfactant-deficient strains of the soil bacteria *P. fluorescens*: AR2, derived from SBW25, and Pf0-2x, derived from Pf0-1 (see Methods). These strains share homologous genetic backgrounds, including highly similar gene regulatory architectures and translated protein products, yet in previous work were observed to evolve divergently[33]. Both engineered strains lack function of the master regulator of flagella-dependent motility, FleQ, and both AR2 and Pf0-2x rapidly re-evolve flagella-mediated motility under strong directional selection[33]. In AR2, this phenotype is achieved in independent lineages via repeatable de novo mutation in the *ntrB* locus of the nitrogen regulatory (ntr) pathway. The parallel evolution of *ntrB* mutants is noteworthy as the locus is consistently targeted, whereas Pf0-2x lines evolve motility via mutations across the ntr regulatory hierarchy[33]. As such parallel evolution between these homologs varies across scale; both are parallel to the phenotype and targeted gene regulatory network, but only one possesses a mutational hotspot that concentrates mutations at a single nucleotide site within a single locus. We conducted a series of experiments to find out why.

Here we show that motility evolves in AR2 in a highly repeatable manner, which is absent in Pf0-2x due to a genetic feature predicated on synonymous variation. The evolution of flagella motility in AR2 is found to target the same nucleotide substitution in over 95% of cases in minimal medium (M9). This outcome is robust across multiple nutrient regimes both in the immotile SBW25 variant (AR2) and another SBW25 variant that is able to access biosurfactant-mediated motility prior to evolution (SBW25 ΔfleQ). The role of selection and the number of viable mutational routes in ensuring the parallel outcome provides some explanation for repeatable evolution to the level of the *ntrB* locus, but not the nucleotide. This, therefore, implies that intra-locus mutational biases are playing a critical role. We reinforce this implication by genetically augmenting the *ntrB* locus to indirectly incriminate mutation bias and reveal a key underlying genetic driver of parallel evolution. Six silent nucleotide changes are introduced within the local region around the frequently targeted site to make AR2's genetic sequence match Pf0-2x, but without altering the protein product. These synonymous changes reduce parallel evolution at the mutational hotspot from >95% to 0%. In a reciprocal experiment, silent changes are introduced to the homologous strain Pf0-2x to match AR2's local native sequence, and these raise parallel evolution at the site from 0% to 80%. These results reveal that local genetic sequence can play a dominant role in ensuring parallel evolutionary outcomes and shine a spotlight on the overlooked mechanistic drivers behind mutational hotspots.

## Results

**SBW25-derived immotile strains evolve motility via highly repeatable evolution.** To quantify the degree of parallel evolution of flagellar motility within the immotile SBW25 model system, we placed 24 independent replicates of AR2 under strong directional selection in a minimal medium environment (M9). Motile mutants were readily identified through emergent motile zones that migrated outward in a concentric circle (Fig. 1A). Clonal samples were isolated from the zone's leading-edge within 24 h of emergence and their genotypes were analysed through either whole-genome sequencing or targeted Sanger sequencing of the *ntrB* locus. Motile strains evolved rapidly (Fig. 1B) and each

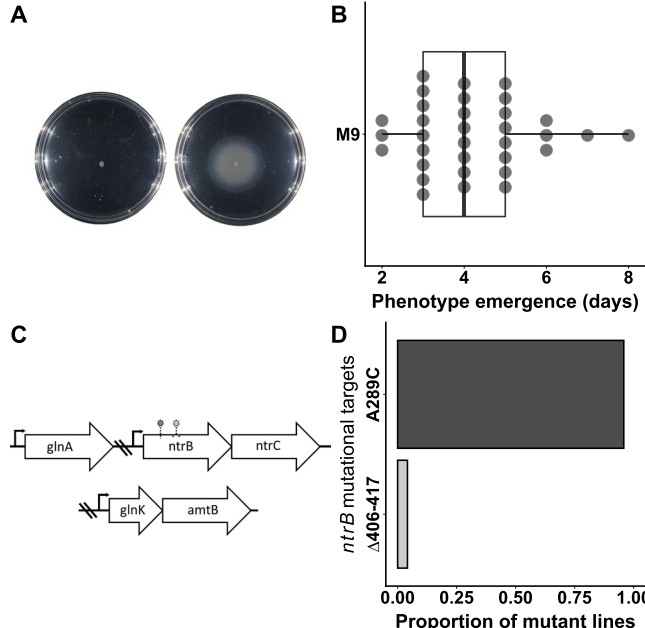

**Fig. 1 Immotile variants of *P. fluorescens* SBW25 (AR2) undergo highly repeatable evolution of flagella-mediated motility. A** Immotile populations evolved on soft agar (left) re-evolved flagella-mediated motility through one-step de novo mutation (right). **B** Phenotype emergence on M9 minimal media appeared rapidly, typically within 3-5 days following inoculation (sample size $N = 33$ independent biological replicates). The centre displays the median, box bounds represent the 25th and 75th percentiles, and whiskers extend from the box bounds to the minima and maxima values, the lengths of which reach up to 1.5 * the inter-quartile range. Individual data points are also plotted. **C** The underlying genetic changes were highly parallel, with all independent lines targeting one of two sites (left circle, A289C and right circle Δ406-417) within the *ntrB* locus at the expense of other sites within the nitrogen (ntr) pathway. **D** A single transversion mutation, A289C, was the most common mutational route, appearing in over 95% of independent lines ($N = 23/24$). Source data are provided as a Source Data file.

independent line was found to be a product of a one-step de novo mutation. All 24 lines had evolved in parallel at the locus level: each had acquired a single, motility-restoring mutation within *ntrB* (Fig. 1C). More surprising, however, was the level of parallel evolution within the locus. 23/24 replicates had acquired a single nucleotide polymorphism at site 289, resulting in a transversion mutation from A to C (hereafter referred to as *ntrB* A289C). This resulted in a T97P missense mutation within NtrB's PAS domain. The remaining sample had acquired a 12-base-pair deletion from nucleotide sites 406–417 (Δ406–417), resulting in an in-frame deletion of residues 136–139 (ΔLVRG) within NtrB's phospho-acceptor domain.

**Repeatable evolution is robust to nutritional environment.** Repeatable evolution could be robust or highly context-dependent, especially when it occurs via de novo mutations with antagonistic pleiotropic effects[34–36]. However, we found that the repeatability of the *ntrB* A289C mutation was robust across all tested conditions, despite evidence of antagonistic pleiotropic effects on growth. We tested for environment-specific antagonistic pleiotropy by measuring relative growth of the ancestral line and both evolved *ntrB* mutants on rich lysogeny broth and minimal medium containing either ammonia as the sole nitrogen source or supplemented with either glutamate (M9 + glu) or glutamine (M9 + gln), both of which are naturally assimilated

and metabolised by the ntr system. Though large fitness costs were evident in M9 minimal medium, supplementing M9 with glu or gln reduced levels of antagonistic pleiotropy for both the *ntrB* A289C and the Δ406–417 mutants (Supplementary Fig. 1). Indeed, the antagonistic pleiotropy of impaired metabolism was sufficiently low in M9 supplemented with the amino acid glutamine (M9 + gln) that motile mutants had increased fitness over the ancestral line in static broth, which was significant in *ntrB* A289C ($P = 0.0361$, Supplementary Fig. 1). These findings show that antagonistic pleiotropy is harsh in M9 and alleviated substantially in other nutritional environments, and therefore evolution in minimal media may have been limiting the viable number of adaptive routes.

We then tested whether repeatable evolution was robust to varying levels of antagonistic pleiotropy in our model system. Our expectation was that supplemented nutrient regimes would lower pleiotropic costs and thus unlock alternative routes of adaptation. We additionally hypothesised that a strain that is able to migrate prior to mutation would also ease starvation-induced selection pressures and could facilitate yet more mutational routes. For this experiment, we therefore utilised an additional immotile variant of SBW25, which unlike AR2 did not have a transposon inserted into *viscB* (see Methods) and thus could migrate via a form of sliding motility prior to mutation (SBW25-Δ*fleQ* (hereafter Δ*fleQ*)[37]. We observed the evolution of a 'blebbing' phenotype (Fig. 1A) in Δ*fleQ* lines despite their ability to migrate in a dendritic fashion; however, we also found that the evolution of blebbing was less frequent under richer nutrient regimes (where populations migrated more rapidly utilising viscosin, see Methods). Overall, there was no evidence that the prevalence of the hotspot mutation *ntrB* A289C changed with nutrient condition (Gene-by-environment interaction: $\chi^2 = 0.875$, df = 3, $P = 0.83$, see Fig. 2). Instead, we observed that the *ntrB* A289C mutation was robust across all tested conditions, featuring in 90–100% of replicates in the Δ*fleQ* strains and 80–100% of replicates in AR2 strains (Fig. 2).

**Repeatable evolution occurs despite motility being accessible via several mutational routes.** Our evolution experiments across nutrient regimes uncovered three novel mutational routes that were observed in a small number of mutants (Fig. 2), revealing that mutational accessibility could not explain the level of observed parallel evolution. Most notable was a non-synonymous A-C transversion mutation at site 683 (*ntrB* A683C) in a Δ*fleQ* line evolved on M9 + gln, resulting in a missense mutation within the NtrB histidine kinase domain. As a single A-C transversion within the same locus, we may expect A683C to mutate at a similar rate to A289C. We also observed a 12 base-pair deletion from sites 410–421 (*ntrB* Δ410–421) in an AR2 line evolved on M9 + gln. Furthermore, we discovered a double mutant in an AR2 line evolved on M9 + glu: one mutation was a single nucleotide deletion at site 84 causing a frameshift within *glnK*, and the second was another A to C transversion at site 688 resulting in a T230P missense mutation within RNA polymerase sigma factor 54.

GlnK is NtrB's native regulatory binding partner and repressor in the ntr pathway, meaning the frameshift mutation alone likely explains the observed motility phenotype. However, as this mutant underwent two independent mutations we will not consider it for the following analysis. In addition, *ntrB* Δ410–421 and *ntrB* Δ406–417, despite targeting different nucleotides, translate into identical protein products (both compress residues LVRGL at positions 136–140 to a single L at position 136). Therefore, we will also group them for the following analysis. Under the assumptions that the three remaining one-step

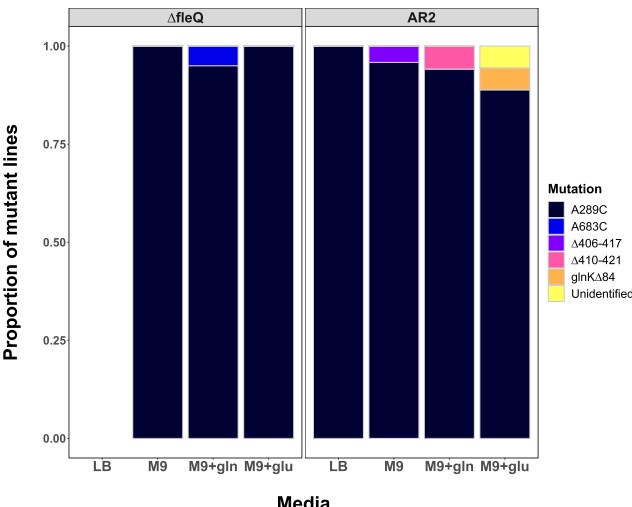

**Fig. 2 Repeatable evolution of mutation *ntrB* A289C is robust across genetic background and nutrient environment (total *N* = 116).** The proportion of each observed mutation is shown on the y axis, and each unique mutational target is highlighted by a colour. Lines were evolved using 4 mM, 8 mM and 16 mM of amino acid supplement glutamate (glu) or glutamine (gln; see Methods). No significant relationship between supplement concentration and evolutionary target was observed (Kruskal–Wallis chi-squared tests, two-sided: AR2 M9 + glu, df = 2, *P* > 0.2; AR2 M9 + gln, df = 1, *P* > 0.23; Δ*fleQ* M9 + gln, df = 1, *P* > 0.3), as such they are grouped visually and for subsequent statistical analysis. *ntrB* mutation A289C was robust across both strain backgrounds (SBW25Δ*fleQ* - shown as Δ*fleQ*, and AR2) and the four tested nutritional environments, remaining the primary target of mutation in all cases (>88%). As such no significant gene-by-environment interaction was observed ($\chi^2$ = 0.875, df = 3, *P* = 0.83). Δ*fleQ* lines evolved on LB were able to migrate rapidly through sliding motility alone, masking any potential emergent flagellate blebs (see[37]). Sample sizes (N) for other categorical variables: Δ*fleQ* – M9: 25, M9 + gln: 20, M9 + glu: 7; AR2 - LB: 5, M9: 24, M9 + gln: 17, M9 + glu: 18. Source data are provided as a Source Data file.

observed mutational routes to novel proteins are (i) equally likely to appear in the population and (ii) equally likely to reach fixation, the original observation of *ntrB* A289C appearing in 23/24 cases becomes exceptional (Bootstrap test: *n* = 1000000, *P* < 1 × 10⁻⁶). The likelihood of our observing this by chance, therefore, is highly unlikely. This means that one or both assumptions are almost certainly incorrect. Either the motility phenotype facilitated by the mutations may be unequal, enabling clonal interference to enforce a repeatable outcome; or the spectrum of adaptive mutations may appear in the population at different rates, resulting in mutation bias. One or both of these elements must be skewing evolution to such a degree that parallel evolution to nucleotide resolution becomes highly predictable.

**Clonal interference cannot explain repeatability to nucleotide resolution.** The adaptationist explanation for parallel evolution is that the observed mutational path is outcompeting all others on their way to fixation. For the purposes of our experiments, we define fixation as establishment on the frontier of the motile zone by the time of sampling. If selection via clonal interference alone was driving repeatable evolutionary outcomes, the superior fitness of the *ntrB* A289C genotype should have allowed it to out-migrate other motile genotypes co-existing in the population. To test if the *ntrB* A289C mutation granted the fittest motility phenotype, we allowed the evolved genotypes (A289C, Δ406–417, A683C and *glnK* Δ84) to migrate independently on the four nutritional backgrounds

and measured their migration area after 48 h. To allow direct comparison, we first engineered the *ntrB* A683C mutation, which originally evolved in the Δ*fleQ* background, into an AR2 strain. We observed that the non-*ntrB* double mutant, *glnK* Δ84, migrated significantly more slowly than *ntrB* A289C in all four nutrient backgrounds (M9: *P* = 0.000708, M9 + gln: *P* = 0.032, M9 + glu: *P* = 0.0025, LB: *P* = 0.0048, Fig. 3). However, *ntrB* A289C did not significantly outperform either of the alternative *ntrB* mutant lines in any environmental condition (*P* value range = 0.074–0.87 Fig. 3). This suggests that selection may have played a role in driving parallel evolution to the level of the *ntrB* locus, but it cannot explain why nucleotide site 289 was so frequently mutated.

To determine if this result remained true when mutant lines were given the opportunity for clonal interference, we directly competed *ntrB* A289C against the alternative *ntrB* mutant lines, Δ406–417 and A683C, on M9 minimal medium. In brief, we co-inoculated the three mutant lines on the same soft agar surface at equal concentrations and allowed them to competitively migrate before sampling from the leading edge after 24 h of competition. Across 15 independent replicates, we observed no significant bias for any *ntrB* mutation at the growth's frontier (*ntrB* A289C = 4/15, *ntrB* Δ406–417 = 8/15, *ntrB* A683 = 3/15; Bootstrap test: *n* = 1000000, *P* > 0.26). We next emulated *ntrB* A289C appearing in the population within a handful of generations after the alternative mutations, and observed that the common genotype is significantly outperformed when inoculated both 6 h and 3 h after the alternative mutant lines (*ntrB* A289C establishment at frontier = 0/16 independent replicates (3 h and 6 h); Bootstrap test: n = 1000000, *P* < 0.005). These results highlight that if motile lines were to appear in the population simultaneously in minimal medium, *ntrB* A289C exhibits no evidence of clonal interference. Furthermore, if the common genotype appears in the population after just a handful of generations of its competitors it fails to establish itself at the frontier. Additionally, given that the range in time before a motility phenotype was observed could vary considerably between independent lines (see above), our data do not support the hypothesis that mutation supply could be high enough to allow multiple phenotype-granting mutations to appear in the population almost simultaneously, as was explored in this assay. As such our evidence suggests that the opportunity for clonal interference during the short course of the experiment would be minimal, and if it were to occur there is no evidence to support it as the causative agent of our repeatable observations of *ntrB* A289C. More likely is that each independent line adhered to the early bird gets the worm maxim, i.e. the *ntrB* mutant which was the first to appear in the population was the genotype subsequently sampled. This, therefore, suggests that the reason *ntrB* A289C is so frequently collected when sampling is owed at least in part to an evolutionary force other than selection and mutational accessibility.

**Silent genetic variation can break a mutational hotspot.** Local mutational biases can play a key role in evolution[24,38]. Such biases can be introduced by changing DNA curvature[26] or through neighbouring tracts of reverse-complement repeats (palindromes and quasi-palindromes), which have been shown to invoke local mutation biases by facilitating the formation of single-stranded DNA hairpins[27]. Therefore we next searched for a local mutation bias at *ntrB* site 289. Previously, we re-evolved motility in two engineered immotile strains of *P. fluorescens*, AR2 (derived from SBW25) and Pf0-2x (derived from Pf0-1)[33]. Although evolved lines in AR2 frequently targeted *ntrB*, Pf0-2x lines fixed mutations across the ntr regulatory pathway. Furthermore, although Pf0-2x did acquire *ntrB* mutations in multiple independent lines, we observed no evidence of *ntrB* site 289 being targeted[33]. The NtrB proteins of SBW25 and Pf0-1 are highly homologous

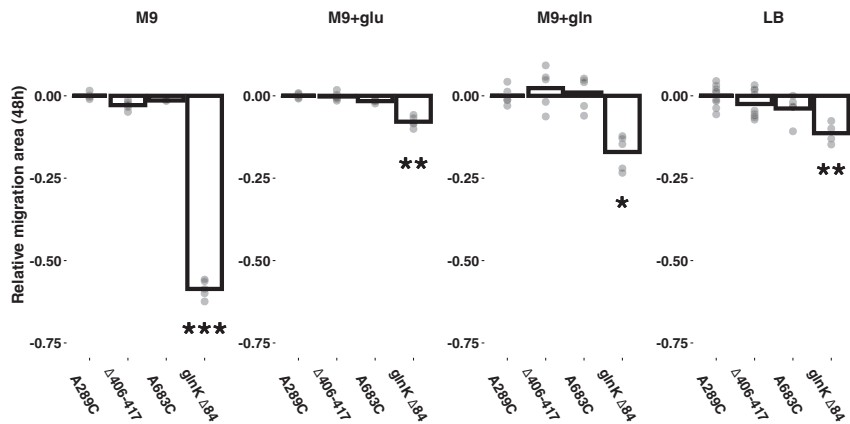

**Fig. 3 *ntrB* mutants possess comparable motility phenotypes.** Surface area of motile zones within an AR2 genetic background following 48 h of growth across four environmental conditions for mutants *ntrB* A289C, *ntrB* Δ406-417, *ntrB* A683C and *glnK* Δ84; *rpoN* A688C. Lysogeny broth (LB, N = 9, 9, 5, 4 respective independent biological replicates), M9 minimal media with ammonia (M9, N = 7, 6, 2, 5), and M9 supplemented with amino acids glutamate (M9 + glu, N = 5, 5, 5, 4) or glutamine (M9 + gln, N = 5, 5, 5, 6) were used as nutritional conditions. Individual data points from biological replicates are plotted and each migration area has been standardised against the surface area of an AR2 *ntrB* A289C mutant grown in the same environment (*ntrB* A289C growth mean = 0). Significance values for *glnK* Δ84; *rpoN* A688C: M9 P = 0.000708, M9 + glu P = 0.0025, M9 + gln P = 0.032, LB P = 0.0048 (Kruskal–Wallis post-hoc Dunn test, one-sided). Source data are provided as a Source Data file.

(95.57% identity) but share less identity at the genetic level (88.88% identity). A considerable portion of this genetic variation is explained by synonymous genetic variation (8.36%) rather than non-synonymous variance (2.76%). Synonymous mutations can play a role in altering local mutation bias. This may occur by altering the nucleotide-triplet to one with a higher mutation rate[25] or by altering the secondary structure of longer DNA tracts via the mechanisms outlined above. Nucleotides that remain unpaired when their neighbouring nucleotides form hairpins with nearby reverse-complement tracts have been observed to exhibit increased mutation rates[39]. Both SBW25 and Pf0-1 were found to have short reverse-complement tracts that flanked site 289, however, the called hairpins were not entirely identical in their composition owing to synonymous variance (Supplementary Fig. 2). Overall, there are 6 synonymous nucleotide substitutions ±5 codons flanking site 289 (C276G, C279T, C285G, C291G, T294G and G300C), which may have been affecting such hairpin formations and impacting local mutation rate.

To test if synonymous sequence was biasing evolutionary outcomes, we replaced the 6 synonymous sites in an AR2 strain with those from a Pf0-1 background (hereafter AR2-sm). Not all these sites formed part of a theoretically predicted stem that overlapped with site 289 but all were targeted due to their close proximity with the site. This ensured that the changes captured any secondary structures forming in the local region around nucleotide position 289. AR2-sm lines were placed under selection for motility and we observed that these lines evolved motility significantly more slowly (Fig. 4A), both in M9 minimal medium and LB (Wilcoxon rank-sum tests with continuity correction: M9, W = 44.5, P < 0.001; LB, W = 22, P < 0.001). Evolved AR2-sm lines that re-evolved motility within 8 days were sampled and their *ntrB* locus was analysed by Sanger sequencing (Fig. 4B). We observed some similar *ntrB* mutations to those identified previously: the *ntrB* A683C mutation was observed in one independent line evolved on LB, and *ntrB* Δ406-417 was also observed in both strain backgrounds. However, the most common genotype of *ntrB* A289C fell from being observed in over 95% of independent lines in M9 to 0%. Furthermore, we observed multiple previously unseen *ntrB* mutations, while a considerable number of lines reported wildtype *ntrB* sequences, instead either targeting another gene of the ntr pathway (*glnK*) or unidentified targets that may lay outside of the network (Fig. 4B).

To test that the A289C transversion remained a viable mutational target in the AR2-sm genetic background, we subsequently engineered the AR2-sm strain with this motility-enabling mutation. We observed that AR2-sm *ntrB* A289C was motile and comparable in phenotype to a *ntrB* A289C mutant that had evolved in the ancestral AR2 genetic background (Supplementary Fig. 3). We additionally found that AR2-sm *ntrB* A289C retained comparable motility to the other *ntrB* mutants evolved from AR2-sm (Supplementary Fig. 3). Therefore, we can determine that the AR2-sm genetic background would not prevent motility following mutation at *ntrB* site 289, nor does it render such a mutation uncompetitive. This, therefore, infers that the sole variable altered between the two strains (the six synonymous changes) are precluding mutation at site 289. Taken together these results strongly suggest that the synonymous sequence immediately surrounding *ntrB* site 289 facilitates its position as a local mutational hotspot, and that local mutational bias is imperative for realising highly parallel evolution in our model system.

**Silent variation can build a mutational hotspot**. As the previous result exemplified the power of synonymous variation in breaking mutational hotspots, we next hypothesised that the same amount of variation could just as readily build a mutational hotspot. To achieve this we engineered a synonymous variant of the immotile Pf0-2x strain (Pf0-2x-sm6). This strain was a reciprocal mutant of AR2-sm, in that it had synonymous variations at the same six sites within *ntrB* but substituted so that they matched AR2's native sequence (G276C, T279C, G285C, G291C, G294T and C300G). We placed both Pf0-2x and Pf0-2x-sm6 under directional selection for motility and observed that Pf0-2x evolved motility slower than Pf0-2x-sm6 (Fig. 5A) and targeted a multitude of sites across multiple loci (Fig. 5B). In stark contrast, Pf0-2x-sm6 evolved both more quickly (Fig. 5A; Wilcoxon rank-sum tests with continuity correction: M9, W = 239.5, P < 0.001; LB, W = 461.5, P < 0.001) and massively more parallel than its native counterpart. Pf0-2x-sm6 fixed *ntrB* A289C in 80% of instances in M9 (8/10 independent lines), despite this de novo mutation not appearing once in a Pf0-2x evolved line (0/22 independent lines, Fig. 5B). The striking differences between the two strains from a Pf0-2x genetic background (Fig. 5) clearly mirror the results observed in the AR2 genetic background (see above). This reveals that a small number of

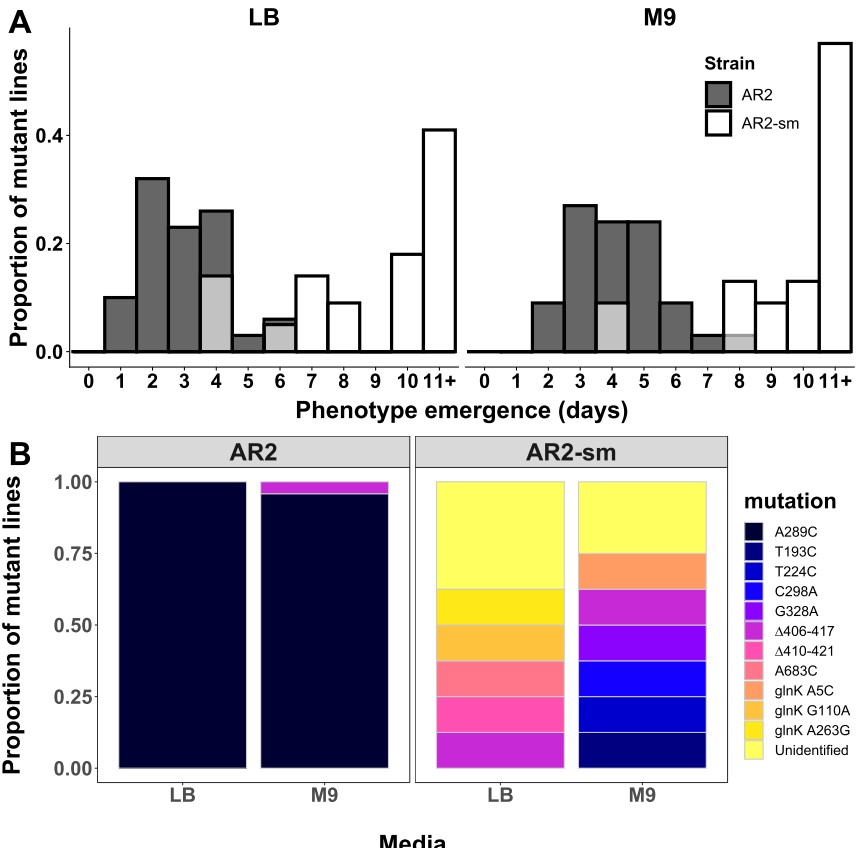

**Fig. 4 A synonymous sequence mutant (AR2-sm) confers a loss of repeatable evolution. A** Histogram of motility phenotype emergence times across independent replicates of immotile SBW25 (AR2, grey) and an AR2 strain with six synonymous substitutions in the *ntrB* locus (AR2-sm, white) in two nutrient conditions. Lines that evolved after the experiment timeframe (10 days) were pooled into bin 11 + . **B** Observed mutational targets following directed evolution of synonymous variants, performed across two environments. Each unique mutation is highlighted by an identifiable colour. (Sample sizes (*N*): AR2: LB *N* = 5, M9 *N* = 24; AR2-sm: LB *N* = 8, M9 *N* = 8). Note that characterised genotypes were sampled within 8 days of experiment start date. Unidentified mutations could not be distinguished from wild type sequences of genes belonging to the nitrogen regulatory pathway (*ntrB, glnK* and *glnA*) which were analysed by Sanger sequencing. *ntrB* Δ406-417 was the only mutational target shared by both lines within the same nutritional environment. Source data are provided as a Source Data file.

synonymous variations can heavily bias mutational outcomes across genetic backgrounds and between homologous strains.

## Discussion

Understanding the evolutionary forces that forge mutational hotspots and repeatedly drive certain mutations to fixation remains an immense challenge. This is true even in simple systems such as the one employed in this study, where clonal bacterial populations were evolved under strong directional selection for very few phenotypes, namely motility and nitrogen metabolism. Here we took immotile variants of *P. fluorescens* SBW25 (AR2) and Pf0-1 (Pf0-2x) that had been observed to repeatedly target the same gene regulatory pathway during the re-evolution of motility[33]. We found that evolving populations of AR2 adapted via de novo substitution mutation in the same locus (*ntrB*) and at the same nucleotide site (A289C) in over 95% of cases in M9 minimal medium. AR2 populations were constrained in which genetic avenues they could take to access the phenotype under selection, but mutational accessibility and clonal interference alone could not explain such a high degree of parallel evolution. Pf0-2x was distinct in that it did not evolve in parallel to nucleotide nor locus resolution. We observed that by introducing synonymous changes around the mutational hotspot (*ntrB* site 289) in both AR2 and Pf0-2x so that their local genetic sequences were swapped, we could push evolving AR2

populations away from the parallel path and pull Pf0-2x lines onto the parallel path. This work reveals that synonymous sequence is an integral factor toward realising highly repeatable evolution and building a mutational hotspot in our system.

More recent studies have revealed that synonymous changes have an underestimated effect on fitness through their perturbances before and during translation. Synonymous sequence variance can impact fitness by changing the stability of mRNA[40–42] and altering codons to perturb or better match the codon-anticodon ratio[43]. We have shown here that synonymous sequence can also be essential for ensuring parallel evolutionary outcomes across genetic backgrounds. Our results strongly infer that this is due to its impact on local mutational biases, which mechanistically may be owed to the formation of single-stranded hairpins that form between short inverted repeats on the same DNA strand[27,44]. The formation of these secondary DNA structures provides a mechanism for intra-locus mutation bias that can operate with local impact and is contingent on DNA sequence variation, as introducing synonymous changes could readily perturb the complementarity of neighbouring inverse repeats (e.g. Supplementary Fig. 2). Furthermore, the finding of just six synonymous mutations having a significant impact on DNA structure would not represent a surprising result, as secondary structures can be altered by single mutations[45].

We can confidently assert that the altered mutational bias is owed to an intra-locus effect, owing to the six synonymous sites

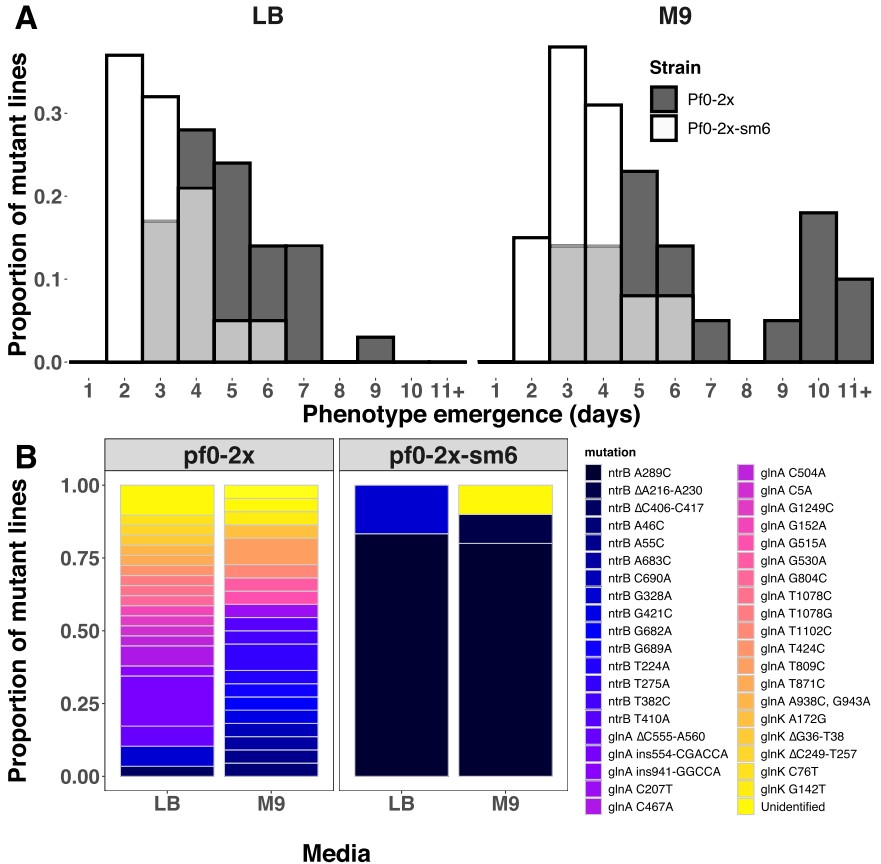

**Fig. 5 A synonymous sequence mutant (Pf0-2x-sm) confers a gain of repeatable evolution. A** Histogram of motility phenotype emergence times across independent replicates of an immotile variant of *P. fluorescens* strain Pf0-1 (Pf0-2x[33],grey) and a Pf0-2x strain with six synonymous substitutions in the *ntrB* locus (Pf0-2x-sm6, white) in two nutrient conditions. **B** Observed mutational targets following directed evolution of synonymous variants, performed across two environments. Each unique mutation is highlighted by an identifiable colour. (Sample sizes (*N*): Pf0-2x: LB *N* = 29, M9 *N* = 22; Pf0-2x-sm: LB *N* = 6, M9 *N* = 10). Unidentified mutations could not be distinguished from wild type sequences of genes belonging to the nitrogen regulatory pathway (*ntrB, glnK* and *glnA*) which were analysed by Sanger sequencing. Mutation *ntrB* A289C was not observed in a single instance in evolved Pf0-2x lines but became the strongly preferred target following synonymous substitution. Source data are provided as a Source Data file.

all residing within 14 bases at either flank of site 289. However, the full elucidation of the secondary structure and genetic mechanistic features enabling this powerful mutation bias awaits further study. We know that at least a portion of the six substituted nucleotide sites is imperative for parallel genetic outcomes, but we do not yet know if other nucleotide features in the local neighbourhood or more broadly e.g. strand orientation[46] or distance from the origin of replication[25] may be combining with local sequence to enforce mutational biases. Interestingly, our data suggest that the mutational hotspot typically mutates so quickly as to mask mutations appearing elsewhere and outside of the nitrogen regulatory pathway, which only appear when the hotspot is perturbed (Figs. 4 and 5). This, therefore, presents the opportunity to additionally quantify the difference in mutation bias owed to secondary structure.

Our findings show that the presence of a mutational hotspot was a stronger deterministic evolutionary force in our system than other variables such as nutrient regime, starvation-induced selection and genetic background. We expected the selective environments to hold some influence over evolutionary outcomes[18] mostly owing to varying levels of antagonistic pleiotropy, which has been found to be a key driver in similar motility studies[8]. Similarly, while parallel evolution can sometimes be impressively robust across genetic backgrounds[47], some innovations are strongly determined by an organism's evolutionary history[48]. Genomic variation also typically combines with

environmental differences to drive populations down diverse paths[49]. However, in our experiments, the strains that share the same 6 synonymous sites evolve more similarly than those that share the same broader genetic background (Figs. 4B and 5B). These results show that strains can share not only high global homology but also similar genomic architecture––including translated protein structures and gene regulatory network organisation––and yet can have strikingly different mutational outcomes when under selection for the exact same traits owing to synonymous variation. This presents intriguing questions as to whether neutral changes could facilitate the dominance of a genotype during adaptation because of a previously acquired mutational hotspot, and asks whether these mutational hotspots can be selectively enforced.

Models looking to describe drivers of adaptive evolution often place precedence on fitness and the number of accessible adaptive routes[50,51] yet pay little attention to local mutational biases (however, see[36]). However, heterogeneity in mutation becomes of paramount importance when systems adhere to the Strong Selection Weak Mutation model (SSWM), which describes instances when an advantageous mutation undergoes a hard sweep to fixation before another beneficial mutation appears[52]. In such cases relative fitness values between adaptive genotypes are relegated to secondary importance behind the likelihood of an adaptive genotype appearing in the population. Indeed, experimental systems that adhere to the SSWM maxim have been

observed to evolve in parallel despite the option of multiple mutational routes to improved fitness[47]. This suggests that uneven mutational biases can be a key driver in forming mutational hotspots and realising parallel evolution, a conclusion which has been reinforced theoretically[24] although empirical data is still lacking. Understanding the mechanistic causes of mutation rate heterogeneity, therefore, will be essential if we are to determine the presence of mutational hotspots that allow for accurate predictions of evolution[38,53]. The challenge remains in identifying what these mechanistic quirks may be, where they may be found, and determining how they impact evolutionary outcomes.

Our work sheds light on the ability of silent genetic variation to build a mutational hotspot with functionally significant evolutionary outcomes. This hotspot is built by an adaptive site under strong directional selection that enjoys biased mutation, facilitating highly repeatable evolution when mutation bias and selection align. Mutation is inherently a random process, but not all sites in the genome possess equal fixation potential. Most changes will not improve a phenotype under selection, and those that do will not necessarily mutate at the same rates. Therefore, we can increase our ability to anticipate the location of a mutational hotspot dramatically, permitted we have a detailed understanding of the evolutionary variables at play. Considerable inroads have already been made toward realising this goal. When searching for adaptive targets, it has been highlighted that loss-of-function mutations are the most frequently observed mutational type under selection[54,55] and that a gene's wider position within its regulatory network determines its propensity in delivering phenotypic change[56]. When searching for mutational biases, it has been shown that parallel evolution at the level of the locus is partially determined by gene length[53] and that molecular apparatus involved in replication and repair can strongly influence the likelihood of a given nucleotide substitution[57,58]. Here, we show that synonymous sequence warrants consideration alongside these other variables by highlighting its impact on the realisation of highly repeatable evolution.

## Methods

**Model system.** Our model system employs strains of the soil microbe *P. fluorescens* SBW25 and Pf0-1 that lack motility through partial gene deletion or disruption of *fleQ*, the master regulator of flagellar motility[37,59]. Motility can be recovered in the absence of FleQ following de novo mutation that allows for the recruitment of a homologous response regulator, of which the most readily targeted is NtrC of the nitrogen regulatory pathway. The initial mutation that facilitates NtrC recruitment occurs in other loci in the nitrogen pathway, resulting in the hyper-phosphorylation of NtrC[33]. Two SBW25-derived strains were used as ancestors in this study: SBW25 Δ*fleQ* (hereafter Δ*fleQ*) and a Δ*fleQ* variant with a functional *viscB* knockout isolated from a transposon library (SBW25Δ*fleQ* IS-ΩKm-hah: PFLU2552, hereafter AR2[37]. Δ*fleQ* can migrate on soft agar (0.25%) prior to mutation via a form of sliding motility, which is owed to the strain's ability to produce viscosin. AR2 cannot produce viscosin and is thus rendered completely immotile prior to mutation. Pf0-1 is a native *gacA* mutant[60] thus does not make viscosin, therefore its Δ*fleQ* variant, Pf0-2x, is rendered completely immotile following disruption of *fleQ*. All cells were grown at 27 °C and all strains used throughout the study (ancestral, evolved and engineered) were stored at −80 °C in 20% glycerol. The nutrient conditions used throughout the work were lysogeny broth (LB) and M9 minimal media containing glucose and 7.5 mM $NH_4$. The minimal media was used in isolation or supplemented with either glutamate (M9 + glu) or glutamine (M9 + gln) at a final supplement concentration of 8 mM unless stated otherwise. A full list of primers used throughout the study can be found in Supplementary Table 1.

**Motility selection experiment.** Immotile variants were placed under selection for flagella-mediated motility using LB and M9 soft agar (0.25%) motility plates. Details of agar preparation are described in[37]. In brief, molten 30 ml aliquots of soft agar were added to 88 mm diameter petri dishes and left to cool at room temperate for four hours. Following this, condensation was removed from the agar surface and lid by drying the plates in a laminar flow hood with the lids removed for 30 minutes. Supplemented concentrations of glutamate (glu)/glutamine (gln) in M9 soft agar were expanded to include final concentrations at 4 mM, 8 mM and 16 mM, as it was observed that biosurfactant-mediated dendritic motility in Δ*fleQ* lines was enhanced at higher supplement concentrations, which masked any emergent blebs. Lowering the gln supplement concentration improved the

likelihood of observing an emergent flagella bleb in M9 + gln motility plates (16 mM: 4/12, 8 mM: 9/20, 4 mM: 7/12 independent lines). However, dendritic motility remained high on all supplements of M9 + glu and persistently masked blebbing (16 mM: 2/12, 8 mM: 3/20, 4 mM: 2/11 independent lines). Although gln/glu supplementation had no bearing on motility in AR2 lines, supplement conditions across both gln/glu were expanded for consistency. Each motility plate was seeded with a single clonal colony derived from a streak plate prepared from clonal cryogenic stock. Initiating the assay with a colony minimised the number of generations from the clonal cryogenic ancestor to the initiation of the assay, helping to ensure a clonal starting population. A single colony was inoculated into the centre of the agar using a sterile pipette tip and monitored daily until the emergence of motile bleb zones (as visualised in Fig. 1A). Samples were isolated from the leading edge, selecting for the strongest motility phenotype on the plate, within 24 h of emergence and streaked onto LB agar (1.5%) to obtain a clonal sample. As Δ*fleQ* lines were motile via dendritic movement prior to re-evolving flagella motility and could visually mask flagella-mediated motile zones, samples were left for 120 h prior to sampling from the leading edge of the growth. An exception was made in instances where blebbing motile zones were observed solely further within the growth area, in which case this area was preferentially sampled.

**Sequencing.** Motility-facilitating changes were determined through PCR amplification and sequencing of *ntrB*, *glnK* and *glnA* genes. Polymerase chain reaction (PCR) products and plasmids were purified using Monarch® PCR & DNA Cleanup Kit (New England Biolabs) and Sanger sequencing was performed by Eurofins Genomics. A subset of AR2 samples evolved on different nutritional backgrounds was additionally screened through Illumina Whole-Genome Sequencing (WGS) by the Milner Genomics Centre and MicrobesNG (LB: $n = 5$, M9: $n = 6$, M9 + gln: $n = 6$, M9 + glu: $n = 7$). This allowed us to screen for potential secondary mutations and to identify rare changes in motile strains with wildtype *ntrB* sequences. We observed no adaptive secondary mutations with motile lines that underwent WGS, however, all AR2-derived strains shared variations from the SBW25 assembly genome at the same 5 positions: 45877 A > AG, 985332 G > GC, 1786536 A > G, 3447980 TCC > T, and 3694384 A > G. The commonality of these mutations strongly indicates that the background AR2 line differs from the reference genome at these positions. We additionally sent 6 Pf0-2x samples with unidentified motility-granting mutations for WGS, and these lines did not share secondary mutations that deviated from the reference genome. *P. fluorescens* SBW25 genome was used as an assembly template for strains derived from this ancestor (NCBI Assembly: ASM922v1, GenBank sequence: AM181176.4) and *P. fluorescens* Pf0-1 was used as an assembly template for Pf0-2x strains (NCBI Assembly: ASM1244v1, GenBank sequence: CP000094.2). Single nucleotide polymorphisms were called using Snippy with default parameters[61] through the Cloud Infrastructure for Microbial Bioinformatics (CLIMB)[62]. In instances where coverage at the called site was low (≤10x), called changes were confirmed by Sanger sequencing.

**Soft agar motility assay.** Cryopreserved samples of AR2 and derived *ntrB* mutants were streaked and grown for 48 h on LB agar (1.5%). Three colonies were then picked, inoculated in LB broth and grown overnight at an agitation of 180 rpm to create biological triplicates for each sample. Overnight cultures were pelleted via centrifugation, their supernatant withdrawn and the cell pellets re-suspended in phosphate buffer saline (PBS) to a final concentration of $OD_{595}$ 1 cells/ml. 1 μl of each replicate was inoculated into soft-agar by piercing the top of the agar with the pipette tip and ejecting the culture into the cavity as the tip was withdrawn. Plates were incubated for 48 h and photographed. Diameters of concentric circle growths were calculated laterally and longitudinally, allowing us to calculate an averaged total surface area using $A = πr^2$ which was then root transformed. This process was repeated as several independent lines underwent a second-step mutation[33] within the 48 h assay. This phenotype was readily observable as a blebbing that appeared at the leading edge along a segment of the circumference, distorting the expected concentric circle of a clonal migrating population. As such these plates were discarded from the study. By completing additional sets of biological triplicates, we ensured that each sample had at least three biological replicates for analysis aside from A683C in the M9 condition, for which two biological replicates are included.

**Invasion assay.** OD-corrected biological triplicates of *ntrB* mutant lines were prepared as outlined above. For each of the biological triplicates, one $OD_{595}$ unit of *ntrB* Δ406–417 and *ntrB* A683C were mixed at equal cell densities (giving two $OD_{595}$ units in total), and one $OD_{595}$ unit of *ntrB* A289C was pelleted and re-suspended in the same volume as the mixed culture. 1 μl of each biological replicate of the re-suspended mixed culture was used to inoculate four soft agar plates as outlined above and incubated, followed by *ntrB* A289C's inoculation into the same cavity after the allotted time had elapsed (0 h, 3 h, and 6 h). When inoculated at 0 h, biological replicates of *ntrB* A289C were added to the plate immediately after *ntrB* Δ406–417 and A683C, with each replicate seeding four soft agar plates. In instances where *ntrB* A289C was added to the plate 3 h or 6 h after *ntrB* Δ406–417 and A683C, overgrowth of culture was avoided by incubating *ntrB* A289C cultures at 22 °C at 0 h until cell pelleting and re-suspension approximately 1 h prior to inoculation. The same 'angle of attack' was used for both instances of inoculation (i.e. the side of the plate that the pipette tip travelled over on its way to the centre),

as small volumes of fluid falling from the tip onto the plate could cause local satellite growth. To avoid the risk of satellite growths affecting results, isolated samples were collected from the leading edge 180° from the angle of attack after a period of 24 h. The *ntrB* locus of one sample per replicate was determined by Sanger sequencing to establish the dominant genotype at the growth frontier.

**Genetic engineering**. A pTS1 plasmid containing *ntrB* A683C was assembled using overlap extension PCR (oePCR) cloning (for detailed protocol see[63]) using vector pTS1 as a template. The *ntrB* synonymous mutants (AR2-sm and Pf0–2x-sm6) and AR2-sm *ntrB* A289C pTS1 plasmids were constructed using oePCR to assemble the insert sequence for allelic exchange, followed by amplification using nested primers and annealed into a pTS1 vector through restriction-ligation. pTS1 is a suicide vector, able to replicate in *E. coli* but not *Pseudomonas*, and contains a tetracycline resistance cassette as well as an open reading frame encoding SacB. Cloned plasmids were introduced to *P. fluorescens* SBW25 strains via puddle mating conjugation with an auxotrophic *E. coli* donor strain ST18. Mutations were incorporated into the genome through two-step allelic exchange, using a method outline by Hmelo et al.[64], with the following adjustments: (i) *P. fluorescens* cells were grown at 27 °C. (ii) An additional passage step was introduced prior to merodiploid selection, whereby colonies consisting of *P. fluorescens* cells that had incorporated the plasmid (merodiploids) were allowed to grow overnight in LB broth free from selection, granting extra generational time for expulsion of the plasmid from the genome. (iii) The overnight cultures were subsequently serially diluted and spot plated onto NSLB agar + 15% (wt/vol) sucrose for AR2 strains and NSLB agar + 5% (wt/vol) sucrose for the Pf0-2x strain. Positive mutant strains were identified through targeted Sanger sequencing of the *ntrB* locus. Merodiploids, which have gone through just one recombination event, will possess both mutant and wild type alleles of the target locus, as well as the *sacB* locus and a tetracycline resistance cassette. However, the wild type allele, *sacB* and tetracycline resistance will be subsequently lost following successful two-step recombination. We therefore also screened these mutant strains for counter-selection escape through PCR-amplification and sequencing of the *sacB* locus and growth on tetracycline. Mutants were only considered successful if there was no product on an agarose gel following amplification of *sacB* alongside appropriate controls, the lines were sensitive to tetracycline, and PCR results of the target locus reported expected changes at the targeted sites.

**Statistics**. All statistical tests and figures were produced in R[65]. Figures were created using the *ggplot* package[66]. Simulated datasets were produced for the Bootstrap tests by randomly drawing from a pool of $n$ values with equal weights $x$ times for one million iterations. Note that for the test examining the mutational spectrum when discussing mutational accessibility, the simulated dataset drew from a pool of three values, and as such encodes that no other mutational routes are possible aside from the observed three. Therefore the derived statistic is an underestimate, with additional routes at any weight lowering the likelihood of repeat observations of a single value. All other tests were completed using functions in base-R aside from the Dunn test, which was performed using the *FSA* package[67]. Along with the Bootstrap tests, the statistical tests used throughout the study were: Kruskal–Wallis chi-squared tests, Kruskal–Wallis post-hoc Dunn test, and Wilcoxon rank-sum tests with continuity correction.

**Reporting summary**. Further information on research design is available in the Nature Research Reporting Summary linked to this article.

## Data availability

Data used for the generation of this manuscript is publicly available and can be accessed [https://doi.org/10.17605/OSF.IO/VUYWP][68]. Source data for Figs. 1, 2, 3, 4, 5, and supplementary figs. 1 and 3 are provided with the paper. Partial sequences in the form of individual loci with mutations can be found on the data repository, and a wiki page that details the filetypes and nomenclature of these partial sequences can be found on the repository homepage. Publicly accessible Pseudomonas sequences were accessed via the Pseudomonas Genome Database (https://www.pseudomonas.com/) and the SBW25 genome assembly was accessed via NCBI (NCBI Assembly: ASM922v1, GenBank sequence: AM181176.4). Whole genomes have been deposited onto GenBank with accession codes: JAIOKC000000000, JAIOKD000000000, JAIOKE000000000, JAIOKF000000000, JAIOKG000000000, JAIOKH000000000, JAIOKN000000000, JAIOKO000000000, JAIOKW000000000. Source data are provided with this paper.

## Code availability

A MATLAB script used for *ntrB* sequence analysis is available on GitHub and can be accessed via the following link: https://github.com/J-S-Horton/Syn-sequence-parallel-evolution. https://doi.org/10.5281/zenodo.5109984[69].

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

## Acknowledgements

We thank Laurence Hurst for comments on earlier versions of this manuscript. In addition, we thank member of the Taylor lab Matthew Shepherd for insightful comments and discussion, and Mark Silby for contributing the ancestral *P. fluorescens* Pf0-2x strain used in the study. This work was supported by the University of Bath University Research Studentship Account (URSA) awarded to TBT and NKP; a Royal Society Dorothy Hodgkin Research Fellowship awarded to TBT (DH150169); and the JABBS Foundation for RWJ. Bioinformatics analysis of the paper was carried out using the Medical Research Council's (MRC) Cloud Infrastructure for Microbial Bioinformatics (CLIMB), and Illumina Whole-Genome Sequencing by the Milner Genomics Centre, Bath, UK and MicrobesNG, Birmingham, UK.

## Author contributions

J.H. and L.F. contributed to data acquisition and analysis. J.H., R.J. and T.T. contributed to project conception and study design. J.H. wrote the manuscript. J.H., R.J., N.P., and T.T. revised the manuscript.

## Competing interests

The authors declare no competing interests.
