## [Peer Review File · Nature Communications]

A mutational hotspot that determines highly repeatable evolution can be built and broken by silent genetic changesREVIEWER COMMENTS

Reviewer #1 (Remarks to the Author):

This paper aims to unpack the genetic causes of repeatable restoration of flagellar motility in two motility-deficient strains of the plant associated bacterium, *Pseudomonas fluorescens*. What governs the repeatable, or parallel, evolution of phenotypes and genotypes is an important subject for evolutionary biology, in part because it lies at the heart of a broader effort to make evolution into a more predictable, rather than retrospective, science. The usual explanation for parallelism is strong selection: repeated evolution of the same trait or genetic change is so unlikely to happen by chance alone that when it does, selection must be driving the process. While in many cases this is an appropriate explanation, it may not be the whole story. Mutation does not occur with equal probability or at equal rates across all sites in a genome, so some sites more likely to contribute to parallelism than others, especially in combination with strong selection. While there is now good evidence that mutational heterogeneity can contribute to parallelism (see Bailey et al. 2017 *BioEssays* 39:1–9), the mechanistic causes remain poorly explored. Enter the work by Horton et al.

The authors build on previous work showing that flagellar motility could be readily and repeatably restored (*Science* 347:1014–1017) by showing that this occurs via distinct genetic ‘solutions’ in two closely related strains – one solution being highly repeatable at the nucleotide level and the other being far more variable. The novel claim of the paper is that this variation in genetic parallelism is controlled by synonymous mutations that change the secondary structure of DNA and, in the process, make some sites more likely to mutate than others.

The evidence they assemble to support their claim is as follows:

1. Restoration of motility in an initially motility-deficient strain is under strong selection caused by local resource depletion on an agar plate, as evidenced by the rapid rate at which motility is restored spontaneously (this is the work by Taylor et al 2015).
2. The variation in genetic parallelism is robust to environmental variation and the extent of antagonistic pleiotropy on other traits. These results mean it is a property of the genotype and not the environment.
3. High degrees of parallelism occur despite alternative routes to motility restoration being possible.
4. Evidence from growth rate assays, as a proxy for fitness, and competition assays among genotypes suggest that there is no intrinsic fitness advantage to the mutation (A289C) most commonly observed. This is an important claim that I will return to below, because I think it is the least well-supported.
5. Reciprocal replacement of 6 synonymous sites flanking the most highly parallel mutation (A289C) in the two strains changes the mutational spectrum dramatically: it reduces parallelism in the strain where it was initially high (AR2) and increases parallelism in the strain where it was initially low (Pf0-2x). This is

a striking and compelling observation that, I feel, constitutes very strong evidence of a role for synonymous variation in governing the mutational spectrum underlying parallelism. The proposed mechanism involves the folding of DNA into stem-loop structures, mutational sites on the loops being more susceptible to mutational change than those in the stem, which pair tightly with a complementary bp. While the authors provide support from computational predictions on secondary structure of their sites, they do not provide additional direct support for this mechanism.

Collectively, the authors have assembled strong evidence that synonymous mutations play a major role in controlling the mutational spectrum available to selection. The weakest dimension to their argument is the data on fitness of the different mutants conferring restored motility (point 4 above) because it does not, contrary to what is claimed in the paper, demonstrate that there is no difference in fitness among the mutants.

Allow me to explain. While the growth rate assays are encouraging, a direct demonstration requires a head-to-head competition like those performed in the co-inoculation experiments reported in Fig 3B. However, the way in which these co-inoculation experiments were performed and analysed cannot be used to support the claim of no fitness difference between the strains. Tracking the number of replicates in which A289C prevailed at the growing front following a period of growth does not estimate fitness and lacks statistical power to make strong claims. The probability that A289C prevails in 3 of 4 replicates, for example, is 0.25 under a null hypothesis that each strain is equally likely to prevail (binomial probability test). Moreover, providing a head start to the other ntrB mutants does not make sense to me. A289C could still be increasing in fitness during the competition but not reach the growing front by the time the assay was completed. The experiment is therefore biased towards finding no differences in fitness.

The alternative, more powerful, design would be to repeat the assay by mixing A289C and the other mutants, perhaps at different frequencies, and then calculate a selection coefficient from the change in relative frequency over time, as is standard practice in experimental evolution. This is a much more sensitive assay that can be done with the same number of replicates. While I understand that it is a lot to ask, especially during a pandemic, to do more experiments, and I don't make this recommendation lightly. However, without this data, I don't think the authors can confidently exclude selection as a driver of parallelism. Either the data needs to be collected or the inferences and conclusions modified to temper their claims somewhat.

That said, I do think there is something here. The reciprocal replacement of synonymous mutations is a compelling result that will be hard to argue with. Bravo. A really fascinating study.

Minor issues:

1. I think there are some conceptual issues that need to be clarified throughout. In particular, clearly distinguishing between the observation of parallelism at the genetic level and what is meant by a mutational hotspot. Simply observing parallelism should not be taken to imply that the site responsible is a mutational hotspot, however defined. A 'hotspot' could be because a site has an unusually high mutation rate relative to neighbouring sites or because it mutates at the same rate but always in the same direction (ie – the mutational spectrum is biased). This needs to be clarified throughout. I would advocate referring to the mutational spectrum versus the mutation rate. I don't think the authors have adequately demonstrated variation in the mutation rate, *sensu stricto*, but they have identified differences in the mutational spectrum after the filtering effect of selection (because they only recover the mutants following restoration of motility).
2. lines 69-70: The authors allude to other mechanisms than selection causing fixation bias. What are these?
3. 'radiate' should be replaced by 'mutate' throughout.
4. Line87-88: clonal interference can be important even on relatively short time scales when founded from an isogenic ancestor in large populations; see Jerison and Desai 2015 for a review (<http://dx.doi.org/10.1016/j.gde.2015.08.008>).
5. Line 95-96: "they evolve divergently due to local genetic differences"; explain
6. Fig 1B – show the original data points as well, to give a visual sense of sample size
7. Line 195 – 'Darwinian explanation' seems historically inaccurate and potentially unfair, as there is nothing inherently non-darwinian about mutational heterogeneity
8. Line 789: what is the mutational index? Some explanation is required.
9. Fig 3B is misleading – connecting the dots implies a time course experiment, which this is not. Moreover, there are multiple *ntxB* mutants labelled on the figure: were the competitions done separately against each one and the results averaged? Or was it a mix of all *ntxB* mutants? This is confusing.

Reviewer #2 (Remarks to the Author):

Review "A mutational hotspot that determines highly repeatable evolution can be built and broken by silent genetic changes"

In the manuscript by Horton et al., the authors investigate the cause of high evolutionary repeatability of motility in the immotile *Pseudomonas fluorescens* strains, AR2 and Pf0-2x. AR2 can evolve flagella-mediated motility in almost all evolving populations through a single SNP (A289C) in *ntrB*. By contrast immotile strain Pf0-2x re-evolves motility through a variety of mutations in the *ntr* pathway. The authors hypothesize that this discrepancy might be caused by a mutational hotspot in AR2. Subsequently to study this, they introduced just six synonymous mutations (matching Pf0-2x sequences) around the A289C mutation of the *ntrB* locus in AR2 which significantly decreases repeatability through this mutation. To that end, the authors conclusively show that a mutational hotspot is responsible for the high degree of repeatability at the nucleotide level in AR2.

This work is very interesting and presents solid data for the presence of -often enigmatic- mutational hotspots. I really liked the hypothesis-driven manner of exploring the remarkable evolutionary repeatability, i.e. showing the presence of several alternative mutational pathways, refuting fixation bias and finally performing experimental evolution of mutation swapped AR2 and Pf0-2x. The methods and statistics are clear, and the results well explained. I support publication of this manuscript in *Nature Communications* as this paper highlights the potential importance of silent variation to (repeatable) evolution.

Specific comments:

How were the cultures grown prior to experimental selection? The authors mention that single clonal colonies were inoculated. Could this pre-growth on agar already result in selection for motile variants and maybe increase the allelic frequency of A289C via selection? What was the effective population size of these colonies?

The authors determined genetic changes mainly by sanger sequencing of candidate genes (*ntrB*, *glnK* and *glnA*). For some AR-2 evolved clones, they did perform WGS. It was not clear to me whether the authors did find any secondary mutations outside of the aforementioned candidate genes. If so, how often or rare were these mutations and could they impact motility (or were they merely hitchhiking).

It wasn't clear to me how the mutation index (Fig S2) was determined. Is this calculated by mfg or quickfold or in another manner?

Reviewer #3 (Remarks to the Author):

In this paper, the authors show that repeatability of evolution in their *Pseudomonas fluorescens* model system is driven by a mutational hotspot that can essentially be turned on and off with a set of 6 synonymous mutations. I found this to be well-written manuscript, presenting some really compelling and unexpected (at least for me) results. I have just a few minor comments/ questions.

I think there is a possible alternate (albeit, much less likely) explanation for the shift in repeatability in the presence of the 6 synonymous mutations. Perhaps the mutational hotspot is maintained in AR2-sm but the synonymous mutations interact with the *ntrB* mutations and so there are epistatic fitness effects driving shifts in fixation rates. I believe that the motile assays aimed at identifying possible differences in motility effects of *ntrB* mutations were all done in the AR2 background (i.e. fig 3). What are the relative effects of these *ntrB* mutations (and others) in the AR2-sm background?

Related to this, please specify the genetic background used in the Fig 3 caption.

Regarding the 6 synonymous mutations – Why were all 6 of these chosen? Was the inferred hairpin structure only impacted when all 6 were modified simultaneously? Or was each one individually inferred to have an impact? I'd like to hear a little more about the choice of mutations here.

Response to Reviewers

Horton et al. (MS reference number NCOMMS-21-13696)

We would firstly like to thank the reviewers for their feedback and numerous positive comments. Their critiques and suggestions have been immensely useful, and we were pleased to implement the proposed changes and strengthen the manuscript. Below we give a point-by-point response to each comment raised.

Reviewer 1:

We were pleased to read that the reviewer enjoyed the rationale and key results outlined in our manuscript and would like to extend our thanks for providing such detailed feedback. They raised many excellent points that, upon addressing, we believe has improved the manuscript. Our amendments following their comments are follows:

Collectively, the authors have assembled strong evidence that synonymous mutations play a major role in controlling the mutational spectrum available to selection. The weakest dimension to their argument is the data on fitness of the different mutants conferring restored motility (point 4 above) because it does not, contrary to what is claimed in the paper, demonstrate that there is no difference in fitness among the mutants ... Tracking the number of replicates in which A289C prevailed at the growing front following a period of growth does not estimate fitness and lacks statistical power to make strong claims. The probability that A289C prevails in 3 of 4 replicates, for example, is 0.25 under a null hypothesis that each strain is equally likely to prevail (binomial probability test). Moreover, providing a head start to the other ntrB mutants does not make sense to me. A289C could still be increasing in fitness during the competition but not reach the growing front by the time the assay was completed. The experiment is therefore biased towards finding no differences in fitness ... Either [alternate] data needs to be collected or the inferences and conclusions modified to temper their claims somewhat.

The reviewer is correct to point out that a claim of even fitness within the manuscript was misleading. As they rightly state, an increase in frequency within the population pool of A289C throughout the course of the experiment would indicate superior fitness, and our assay was not performed to capture this information. The intent of the assay was instead to address the argument that the introduced artificial bottleneck (i.e. sampling from the leading edge) was facilitating clonal interference during the brief window from the appearance of motility to sampling (≤ 24 h) and therefore biasing the sampled genotype. As such the assay was designed not to measure fitness, but to emulate the original evolution experiment and determine if A289C could establish itself on the frontier if it appeared in the population concurrently and after alternate ntrB mutations. We have now updated the main text [lines 85-87, 189, 193-197] to remove misleading comments regarding fitness and add clarity to the intention of the assay by switching the focus to clonal interference. In addition, we have expanded the assay to include all 3 ntrB mutants and increased the number of replicates to increase the statistical power of our observations. See [lines 209-228] for a description of and the conclusions drawn from amended assay. Additionally [lines 461-469, 504-505] in the materials and methods have been updated to describe the amended assay.

Minor issues

1. I think there are some conceptual issues that need to be clarified throughout. In particular, clearly distinguishing between the observation of parallelism at the genetic level and what is

meant by a mutational hotspot. Simply observing parallelism should not be taken to imply that the site responsible is a mutational hotspot, however defined. A 'hotspot' could be because a site has an unusually high mutation rate relative to neighbouring sites or because it mutates at the same rate but always in the same direction (ie – the mutational spectrum is biased). This needs to be clarified throughout. I would advocate referring to the mutational spectrum versus the mutation rate. I don't think the authors have adequately demonstrated variation in the mutation rate, *sensu stricto*, but they have identified differences in the mutational spectrum after the filtering effect of selection (because they only recover the mutants following restoration of motility).

We agree that the study doesn't discriminate between elevated rate and a biased spectrum. We have now updated [lines 70-73, 109-110, 189-190, 337, 357, 371, 780] to remove referral to mutation rate and instead emphasise that the focus of the study is on mutation bias.

2. lines 69-70: The authors allude to other mechanisms than selection causing fixation bias. What are these?

[Lines 65-67] have been updated to expand on the description of the referenced paper.

3. 'radiate' should be replaced by 'mutate' throughout.

[Lines 71, 208, 277, 778, 780] have been updated to replace 'radiate' with 'mutate'.

4. Line 87-88: clonal interference can be important even on relatively short time scales when founded from an isogenic ancestor in large populations; see Jerison and Desai 2015 for a review (<http://dx.doi.org/10.1016/j.gde.2015.08.008>).

In relation to the point regarding our amended assay measuring clonal interference, [lines 85-87] have been updated to introduce this problem and this paper has been included as a reference.

5. Line 95-96: "they evolve divergently due to local genetic differences"; explain

The sentence has been adjusted to remove "due to local genetic differences" [line 92]. This comment was made in reference to the impact of synonymous changes, but as this finding is not properly introduced until the next paragraph, the words have been removed for clarity.

6. Fig 1B – show the original data points as well, to give a visual sense of sample size

The figure has been updated to include individual data points, which have been plotted over the boxplot.

7. Line 195 – 'Darwinian explanation' seems historically inaccurate and potentially unfair, as there is nothing inherently non-darwinian about mutational heterogeneity

[Line 194] – switched to "adaptationist"

8. Line 789: what is the mutational index? Some explanation is required.

[Lines 770-773] have now been added to describe how the Mutational Index is calculated:

"The stability, structure and included nucleotide tracts of the predicted hairpins differ between strains and determine the mutated nucleotide site's Mutational Index (MI), which is a multiplication of the secondary structure's maximum energy (ΔG) and the percentage of alternative DNA folds in which the base of interest is unpaired."

9. Fig 3B is misleading – connecting the dots implies a time course experiment, which this is not. Moreover, there are multiple ntrB mutants labelled on the figure: were the competitions done separately against each one and the results averaged? Or was it a mix of all ntrB mutants? This is confusing.

As outlined in our reply to the reviewer's major comment we have now included a description of an amended assay in the main text, and this will stand in place of the assay used for Fig. 3B. The description of the expanded assay can be found on [lines 209-228] and the updated materials and methods found on [lines 461-469, 504-505]. We agree with the reviewer that visualising the data with connected points suggests a time course experiment. Furthermore, an alternative presentation – such as independent stacked bars – would remove the issue of a time-series appearance but risks instead suggesting that the assays addressed the relative density of genotypes in the population. As such, we have decided to reduce figure 3 so that just the independent motility phenotypes (previously Fig. 3A) is visualised, and the amended assay is described in the text. We feel this will ensure clarity for the reader.

Reviewer 2:

We thank the reviewer for supporting publication of the manuscript and for their positive words regarding the manuscript's narrative structure. The reviewer also highlighted some areas where extra clarity is needed in the text, and as such we have updated the manuscript as follows:

How were the cultures grown prior to experimental selection? The authors mention that single clonal colonies were inoculated. Could this pre-growth on agar already result in selection for motile variants and maybe increase the allelic frequency of A289C via selection? What was the effective population size of these colonies?

The colonies used for the evolution experiments were sourced from a streak plate which had been grown directly from a clonal cryogenic ancestral stock. The main advantage of this approach is that it minimises the number of generations from the clonal cryogenic ancestor to the initiation of the assay, helping to ensure a clonal starting population. Furthermore there is minimal directional selection operating on separated cells growing on solid agar, and under this condition motility-causing mutations would be maladaptive (owing to the pleiotropic effects of mutation, as described in [lines 138-152]). The main consequence of not correcting for starting population size with regards to this experiment is that it may impact the likelihood of observing clonal interference, especially when the starting population size is high. We have now addressed the role of clonal interference in greater detail [lines 209-228] and have added extra detail to the materials and methods [lines 417-420] to outline our rationale for initiating the assay with colonies.

The authors determined genetic changes mainly by sanger sequencing of candidate genes (ntrB, glnK and glnA). For some AR-2 evolved clones, they did perform WGS. It was not clear to me whether the authors did find any secondary mutations outside of the aforementioned candidate genes. If so, how often or rare were these mutations and could they impact motility (or were they merely hitchhiking).

We thank the reviewer for bringing this to our attention. In [lines 435-436] of the submitted manuscript we refer to searching for secondary mutations but do not report on whether any were found. We have now added additional description [lines 436-440] to the materials and methods to

clarify that no independent secondary mutations were found, but that some background mutations exist in the AR2 ancestral strain that differ from the whole genome assembly of SBW25.

It wasn't clear to me how the mutation index (Fig S2) was determined. Is this calculated by mfg or quickfold or in another manner?

[Lines 770-773] have now been added to the figure legend of Supplementary Figure 2 to describe how the Mutational Index is calculated.

Reviewer 3:

We were very pleased to read that the reviewer enjoyed both the results and the writing style of the manuscript. The reviewer also highlighted some important questions that required extra clarity in the text. These are as follows:

I think there is a possible alternate (albeit, much less likely) explanation for the shift in repeatability in the presence of the 6 synonymous mutations. Perhaps the mutational hotspot is maintained in AR2-sm but the synonymous mutations interact with the ntrB mutations and so there are epistatic fitness effects driving shifts in fixation rates. I believe that the motile assays aimed at identifying possible differences in motility effects of ntrB mutations were all done in the AR2 background (i.e. fig 3). What are the relative effects of these ntrB mutations (and others) in the AR2-sm background?

This reviewer is correct to highlight that as the sm genetic background is locally different to AR2, epistasis may well have been playing a role in negating fixation following mutation at the hotspot site. We addressed this in the manuscript by engineering an AR2-sm line with the mutation A289C and measuring its motility phenotype (supplementary Fig. 3) versus the AR2 ancestor and other collected AR2-sm ntrB lines (which were sm counterparts of the AR2 motile lines). We observed that A289C-sm was at the very least comparable in fitness to its ancestor and the alternate motile lines [lines 269-280], and so are confident that the synonymous mutations do not carry any epistatic effects.

Related to this, please specify the genetic background used in the Fig 3 caption.

[Lines 699 and 701] have been updated in the figure legend to stipulate the genetic background used for the assay.

Regarding the 6 synonymous mutations – Why were all 6 of these chosen? Was the inferred hairpin structure only impacted when all 6 were modified simultaneously? Or was each one individually inferred to have an impact? I'd like to hear a little more about the choice of mutations here.

[Lines 255-258] have been updated to provide additional explanation of why these sites were selected. The text reads as follows: "Not all these sites formed part of a theoretically predicted stem that overlapped with site 289, but all were targeted due to their close proximity with the site. This ensured that the changes captured any secondary structures forming in the local region around nucleotide position 289".

Additional edits:

[Line 355] – Replaced citation from unpublished archive (Krug, 2019) with published manuscript (Zagorski et al., 2016).

[Line 360] – Clarification of SSWM model.

Corrected mislabelling of rare snps and characterised some previously unidentified lines in Fig. 4B and Fig. 5B.

Figures have been updated to vector format.

REVIEWER COMMENTS

Reviewer #1 (Remarks to the Author):

This paper aims to unpack the genetic causes of repeatable restoration of flagellar motility in two motility-deficient strains of the plant associated bacterium, *Pseudomonas fluorescens*. What governs the repeatable, or parallel, evolution of phenotypes and genotypes is an important subject for evolutionary biology, in part because it lies at the heart of a broader effort to make evolution into a more predictable, rather than retrospective, science. The usual explanation for parallelism is strong selection: repeated evolution of the same trait or genetic change is so unlikely to happen by chance alone that when it does, selection must be driving the process. While in many cases this is an appropriate explanation, it may not be the whole story. Mutation does not occur with equal probability or at equal rates across all sites in a genome, so some sites more likely to contribute to parallelism than others, especially in combination with strong selection. While there is now good evidence that mutational heterogeneity can contribute to parallelism (see Bailey et al. 2017 *BioEssays* 39:1–9), the mechanistic causes remain poorly explored. Enter the work by Horton et al.

The authors build on previous work showing that flagellar motility could be readily and repeatably restored (*Science* 347:1014–1017) by showing that this occurs via distinct genetic ‘solutions’ in two closely related strains – one solution being highly repeatable at the nucleotide level and the other being far more variable. The novel claim of the paper is that this variation in genetic parallelism is controlled by synonymous mutations that change the secondary structure of DNA and, in the process, make some sites more likely to mutate than others.

The evidence they assemble to support their claim is as follows:

1. Restoration of motility in an initially motility-deficient strain is under strong selection caused by local resource depletion on an agar plate, as evidenced by the rapid rate at which motility is restored spontaneously (this is the work by Taylor et al 2015).
2. The variation in genetic parallelism is robust to environmental variation and the extent of antagonistic pleiotropy on other traits. These results mean it is a property of the genotype and not the environment.
3. High degrees of parallelism occur despite alternative routes to motility restoration being possible.
4. Evidence from growth rate assays, as a proxy for fitness, and competition assays among genotypes suggest that there is no intrinsic fitness advantage to the mutation (A289C) most commonly observed. This is an important claim that I will return to below, because I think it is the least well-supported.

5. Reciprocal replacement of 6 synonymous sites flanking the most highly parallel mutation (A289C) in the two strains changes the mutational spectrum dramatically: it reduces parallelism in the strain where it was initially high (AR2) and increases parallelism in the strain where it was initially low (Pf0-2x). This is a striking and compelling observation that, I feel, constitutes very strong evidence of a role for synonymous variation in governing the mutational spectrum underlying parallelism. The proposed mechanism involves the folding of DNA into stem-loop structures, mutational sites on the loops being more susceptible to mutational change than those in the stem, which pair tightly with a complementary bp. While the authors provide support from computational predictions on secondary structure of their sites, they do not provide additional direct support for this mechanism.

Collectively, the authors have assembled strong evidence that synonymous mutations play a major role in controlling the mutational spectrum available to selection. The weakest dimension to their argument is the data on fitness of the different mutants conferring restored motility (point 4 above) because it does not, contrary to what is claimed in the paper, demonstrate that there is no difference in fitness among the mutants.

Allow me to explain. While the growth rate assays are encouraging, a direct demonstration requires a head-to-head competition like those performed in the co-inoculation experiments reported in Fig 3B. However, the way in which these co-inoculation experiments were performed and analysed cannot be used to support the claim of no fitness difference between the strains. Tracking the number of replicates in which A289C prevailed at the growing front following a period of growth does not estimate fitness and lacks statistical power to make strong claims. The probability that A289C prevails in 3 of 4 replicates, for example, is 0.25 under a null hypothesis that each strain is equally likely to prevail (binomial probability test). Moreover, providing a head start to the other *ntrB* mutants does not make sense to me. A289C could still be increasing in fitness during the competition but not reach the growing front by the time the assay was completed. The experiment is therefore biased towards finding no differences in fitness.

The alternative, more powerful, design would be to repeat the assay by mixing A289C and the other mutants, perhaps at different frequencies, and then calculate a selection coefficient from the change in relative frequency over time, as is standard practice in experimental evolution. This is a much more sensitive assay that can be done with the same number of replicates. While I understand that it is a lot to ask, especially during a pandemic, to do more experiments, and I don't make this recommendation lightly. However, without this data, I don't think the authors can confidently exclude selection as a driver of parallelism. Either the data needs to be collected or the inferences and conclusions modified to temper their claims somewhat.

That said, I do think there is something here. The reciprocal replacement of synonymous mutations is a compelling result that will be hard to argue with. Bravo. A really fascinating study.

Minor issues:

1. I think there are some conceptual issues that need to be clarified throughout. In particular, clearly distinguishing between the observation of parallelism at the genetic level and what is meant by a mutational hotspot. Simply observing parallelism should not be taken to imply that the site responsible is a mutational hotspot, however defined. A 'hotspot' could be because a site has an unusually high mutation rate relative to neighbouring sites or because it mutates at the same rate but always in the same direction (ie – the mutational spectrum is biased). This needs to be clarified throughout. I would advocate referring to the mutational spectrum versus the mutation rate. I don't think the authors have adequately demonstrated variation in the mutation rate, *sensu stricto*, but

they have identified differences in the mutational spectrum after the filtering effect of selection (because they only recover the mutants following restoration of motility).

2. lines 69-70: The authors allude to other mechanisms than selection causing fixation bias. What are these?

3. 'radiate' should be replaced by 'mutate' throughout.

4. Line 87-88: clonal interference can be important even on relatively short time scales when founded from an isogenic ancestor in large populations; see Jerison and Desai 2015 for a review (<http://dx.doi.org/10.1016/j.gde.2015.08.008>).

5. Line 95-96: "they evolve divergently due to local genetic differences"; explain

6. Fig 1B – show the original data points as well, to give a visual sense of sample size

7. Line 195 – 'Darwinian explanation' seems historically inaccurate and potentially unfair, as there is nothing inherently non-darwinian about mutational heterogeneity

8. Line 789: what is the mutational index? Some explanation is required.

9. Fig 3B is misleading – connecting the dots implies a time course experiment, which this is not. Moreover, there are multiple ntrB mutants labelled on the figure: were the competitions done separately against each one and the results averaged? Or was it a mix of all ntrB mutants? This is confusing.

Reviewer #2 (Remarks to the Author):

Review "A mutational hotspot that determines highly repeatable evolution can be built and broken by silent genetic changes"

In the manuscript by Horton et al., the authors investigate the cause of high evolutionary repeatability of motility in the immotile *Pseudomonas fluorescens* strains, AR2 and Pf0-2x. AR2 can evolve flagella-mediated motility in almost all evolving populations through a single SNP (A289C) in ntrB. By contrast immotile strain Pf0-2x re-evolves motility through a variety of mutations in the ntr pathway. The authors hypothesize that this discrepancy might be caused by a mutational hotspot in AR2. Subsequently to study this, they introduced just six synonymous mutations (matching Pf0-2x sequences) around the A289C mutation of the ntrB locus in AR2 which significantly decreases repeatability through this mutation. To that end, the authors conclusively show that a mutational hotspot is responsible for the high degree of repeatability at the nucleotide level in AR2.

This work is very interesting and presents solid data for the presence of -often enigmatic- mutational hotspots. I really liked the hypothesis-driven manner of exploring the remarkable evolutionary repeatability, i.e. showing the presence of several alternative mutational pathways, refuting fixation bias and finally performing experimental evolution of mutation swapped AR2 and Pf0-2x. The methods and statistics are clear, and the results well explained. I support publication of this manuscript in Nature Communications as this paper highlights the potential importance of silent variation to (repeatable) evolution.

Specific comments:

How were the cultures grown prior to experimental selection? The authors mention that single clonal colonies were inoculated. Could this pre-growth on agar already result in selection for motile variants and maybe increase the allelic frequency of A289C via selection? What was the effective population size of these colonies?

The authors determined genetic changes mainly by sanger sequencing of candidate genes (*ntrB*, *glnK* and *glnA*). For some AR-2 evolved clones, they did perform WGS. It was not clear to me whether the authors did find any secondary mutations outside of the aforementioned candidate genes. If so, how often or rare were these mutations and could they impact motility (or were they merely hitchhiking).

It wasn't clear to me how the mutation index (Fig S2) was determined. Is this calculated by *mfg* or quickfold or in another manner?

Reviewer #3 (Remarks to the Author):

In this paper, the authors show that repeatability of evolution in their *Pseudomonas fluorescens* model system is driven by a mutational hotspot that can essentially be turned on and off with a set of 6 synonymous mutations. I found this to be well-written manuscript, presenting some really compelling and unexpected (at least for me) results. I have just a few minor comments/ questions.

I think there is a possible alternate (albeit, much less likely) explanation for the shift in repeatability in the presence of the 6 synonymous mutations. Perhaps the mutational hotspot is maintained in AR2-sm but the synonymous mutations interact with the *ntrB* mutations and so there are epistatic fitness effects driving shifts in fixation rates. I believe that the motile assays aimed at identifying possible differences in motility effects of *ntrB* mutations were all done in the AR2 background (i.e. fig 3). What are the relative effects of these *ntrB* mutations (and others) in the AR2-sm background?

Related to this, please specify the genetic background used in the Fig 3 caption.

Regarding the 6 synonymous mutations – Why were all 6 of these chosen? Was the inferred hairpin structure only impacted when all 6 were modified simultaneously? Or was each one individually inferred to have an impact? I'd like to hear a little more about the choice of mutations here.

REVIEWERS' COMMENTS

Reviewer #1 (Remarks to the Author):

The revised manuscript by Horton et al. is much improved. I appreciate the effort they authors have taken in addressing my criticism regarding the fitness assays among distinct motility-restoring mutants. This section reads far more clearly than before and is, in my view, provides sufficient evidence to show that variation in fitness effects are not driving the fixation of one mutant over others. Moreover, the authors have improved the readability and clarity of their manuscript greatly.

One small clarification is required: on line 81 the authors state that clonal interference can become important when mutation rate is high relative to selection. It is more accurate to state that clonal interference is important when mutation *supply* rate (the product of effective population size and mutation rate) is high relative to selection. High mutation supply rates can result either because of high mutation rates, as stated, or large effective population size. Clonal interference occurs when $NU > 1$.

Reviewer #2 (Remarks to the Author):

The authors have satisfactorily addressed my comments in their response and in the manuscript. From my point of view, the manuscript is ready for publication in Nature Communications. I look forward to seeing this paper published.

Reviewer #3 (Remarks to the Author):

The authors have satisfactorily clarified and addressed my previous questions and comments. I also really appreciate the expanded exploration and clarification of how clonal interference might (but it turns out doesn't) play a role in which motility mutations fix (in response to another reviewer's comments). This is a really nice study.

Author response to reviewer's comments

We thank all reviewers for their positive feedback and for supporting publication of the manuscript. Reviewers 2 and 3 requested no further clarifications or adjustments, and Reviewer 1 suggested one clarification as follows:

“One small clarification is required: on line 81 the authors state that clonal interference can be come important when mutation rate is high relative to selection. It is more accurate to state that clonal interference is important when mutation *supply* rate (the product of effective population size and mutation rate) is high relative to selection. High mutation supply rates can result either because of high mutation rates, as stated, or large effective population size. Clonal interference occurs when $NU > 1$.”

We thank the reviewer for this clarification and have now amended line 81 from “when mutation rate” to “when the rate of mutation supply”.

Reviewer's comments

Reviewer #1 (Remarks to the Author):

The revised manuscript by Horton et al. is much improved. I appreciate the effort they authors have taken in addressing my criticism regarding the fitness assays among distinct motility-restoring mutants. This section reads far more clearly than before and is, in my view, provides sufficient evidence to show that variation in fitness effects are not driving the fixation of one mutant over others. Moreover, the authors have improved the readability and clarity of their manuscript greatly.

One small clarification is required: on line 81 the authors state that clonal interference can be come important when mutation rate is high relative to selection. It is more accurate to state that clonal interference is important when mutation *supply* rate (the product of effective population size and mutation rate) is high relative to selection. High mutation supply rates can result either because of high mutation rates, as stated, or large effective population size. Clonal interference occurs when $NU > 1$.

Reviewer #2 (Remarks to the Author):

The authors have satisfactorily addressed my comments in their response and in the manuscript. From my point of view, the manuscript is ready for publication in Nature Communications. I look forward to seeing this paper published.

Reviewer #3 (Remarks to the Author):

The authors have satisfactorily clarified and addressed my previous questions and comments. I also really appreciate the expanded exploration and clarification of how clonal interference might (but it turns out doesn't) play a role in which motility mutations fix (in response to another reviewer's comments). This is a really nice study.